



# Role of vegetation in representing land surface temperature in the CHTESSEL (CY45R1) and SURFEX-ISBA (v8.1) land surface models: a case study over Iberia

5 **Miguel Nogueira[1], Clément Albergel[2], Souhail Boussetta[3], Frederico Johannsen[1], Isabel F. Trigo[4], Sofia L. Ermida[4], João P. A. Martins[4], Emanuel Dutra[1]**

1 Instituto Dom Luiz, IDL, Faculty of Sciences, University of Lisbon, 1749-016 Lisbon, Portugal;

2 CNRM, Université de Toulouse, Meteo-France, CNRS, Toulouse, France;

ECMWF, Reading, UK;

4 Instituto Português do Mar e da Atmosfera, 1749-077 Lisboa, Portugal

Correspondence to: Miguel Nogueira (mdnogueira@fc.ul.pt)

**Abstract.** Earth observations were used to evaluate the representation of Land Surface Temperature (LST) and vegetation coverage over Iberia in two state-of-the-art land surface models (LSMs) - the European Centre for Medium
Range Weather Forecasting (ECMWF) Carbon-Hydrology Tiled ECMWF Scheme for Surface Exchanges over Land (CHTESSEL) and the Météo-France Interaction between Soil Biosphere and Atmosphere model (ISBA) within the SURface EXternalisée modelling platform (SURFEX-ISBA) for the 2004-2015 period. The results show that the daily maximum LST simulated by CHTESSEL over Iberia is affected by a large cold bias during summer months when compared against the Satellite Application Facility on Land Surface Analysis (LSA-SAF), reaching magnitudes larger
than 10ºC over wide portions of central and southwestern Iberia. This error is shown to be tightly linked to a misrepresentation of the vegetation cover. In contrast, SURFEX simulations did not display such a cold bias. We show that this was due to the better representation of vegetation cover in SURFEX, which uses an updated land cover dataset (ECOCLIMAP-II) and an interactive vegetation evolution, representing seasonality. The representation of vegetation over Iberia in CHTESSEL was improved by combining information from the European Space Agency
Climate Change Initiative (ESA-CCI) land cover dataset with the Copernicus Global Land Service (CGLS) Leaf Area Index (LAI) and fraction of vegetation coverage (FCOVER). The proposed improvement in vegetation also includes a clumping approach that introduces seasonality to the vegetation cover. The results show significant added value, removing the daily maximum LST summer cold bias completely, without reducing the accuracy of the simulated LST, regardless of season or time of the day. The striking performance differences between SURFEX and CHTESSEL were
fundamental to guide the developments in CHTESSEL highlighting the importance of using different models. This work has important implications: first, it takes advantage of LST, a key variable in surface-atmosphere energy and water exchanges, which is closely related to satellite top-of-atmosphere observations, to improve model's representation of land surface processes. Second, CHTESSEL is the land surface model employed by ECMWF in the production of their weather forecasts and reanalysis, hence systematic errors in land surface variables and fluxes are
then propagated into those products. Indeed, we show that the summer daily maximum LST cold bias over Iberia in CHTESSEL is present in the widely used ECMWF fifth generation reanalysis (ERA5). Finally, our results provide hints into the interaction between vegetation land-atmosphere exchanges, highlighting the relevance of the vegetation cover and respective seasonality in representing land surface temperature in both CHTESSEL and SURFEX. As a whole, this work demonstrates the added value in using multiple earth observation products for constraining and
improving weather and climate simulations.



## 1. Introduction


Land surface temperature (LST) plays a central role in the land-atmosphere energy, water and carbon exchanges. Specifically, the LST is a key variable for the emission of long-wave radiation by the surface. Additionally, the LST modulates the heat exchanges with the underlying soil layer and overlying atmosphere via the turbulent fluxes, affecting directly and indirectly (via soil water) plant growth. Given LST crucial role for the Earth's climate system,

it has been considered as an Essential Climate Variable into the Global Climate Observing System (Bojinski et al., 2014)

Currently, remote sensing techniques represent the best means to estimate land surface properties with adequate temporal and spatial sampling, ensuring a wide spatial coverage at high resolution. LST may be estimated from remote sensing observations as the directional radiometric temperature of the surface (Norman and Becker, 1995). Satellite-

based LST estimates are often derived from the outgoing thermal infrared radiation (TIR) measured at the top-of-atmosphere (TOA). This spectral band (corresponding to the 8-13 mm range) is particularly appropriate as it presents relatively weak atmospheric attenuation under clear sky conditions and includes the peak of the Earth's spectral radiance (Li et al., 2013; Ermida et al., 2019). However, estimation of LST from TIR at TOA still requires correcting for the atmospheric attenuation along the path and the reflection of downward radiance. This is a challenging task

which requires detailed knowledge of the 3-dimensional atmospheric structure at each instant (including temperature, water vapor concentration, clouds and aerosol load). Additionally, satellite-based estimates of LST also require detailed knowledge on land surface emissivity, while the directional character of LST satellite products should be kept in mind, especially over heterogeneous and non-isothermal surfaces (e.g., Trigo et al., 2011; Ermida et al., 2018). Furthermore, LST estimates derived from TIR are limited to clear sky observation, representing a significant limitation

to its coverage (e.g., Trigo et al., 2011; Li et al., 2013; Ermida et al., 2019). LST estimates may also be derived from satellite passive microwave (MW) measurements. Although this method has the main advantage of allowing LST estimates under all-weather conditions, MW observations introduce other sources of uncertainty, which are not present in TIR-based estimates. Given the higher penetration depth of MW radiation, LST estimated from MW measurements may correspond to subsurface temperature, which differs from the skin temperature estimated from TIR

measurements. Furthermore, penetration depth varies with soil/vegetation type and also with soil moisture, which makes it difficult to convert MW- based LST into TIR-based (or skin) LST (Ermida et al., 2017). MW LST estimates also have usually lower spatial resolution, and lower accuracy values, typically in the 5-6 K range (e.g., Aires et al., 2001; Prigent et al., 2016; Duan et al., 2017), when compared with TIR LST (with uncertainties in the 1-2 K; Trigo et al., 2011)

LST estimates may also be obtained from Land surface model (LSM) simulations. These models are physically based, consistent throughout the entire simulation period (while satellite-based records are affected by the decay and replacement of the instruments), available under all sky conditions, and not limited to the satellite-period (they can be extended into the past or future, given the appropriate forcing). However, the model-based approach is affected by uncertainties in the model formulation due to the limited grid resolution and incomplete knowledge of the wide range

of physical processes involved. Additionally, model LST values are also affected by uncertainties in the atmospheric conditions and incoming surface radiative fluxes, provided as input to LSMs, and in the surface properties represented in the model (e.g. vegetation coverage, albedo, roughness lengths, etc.). Hence, it is not surprising that several previous studies have reported significant errors in LST estimated by LSMs (e.g. Garand, 2003; Mitchell et al., 2004; Edwards, 2010; Zheng et al., 2012; Scarino et al., 2013; Wang et al., 2014; Trigo et al., 2015; Orth et al., 2017; Zhou et al.,

2017; Johansson et al., 2019). In fact, in the case of simulating latent and sensible heat fluxes, previous works have shown that physically based LSMs can be outperformed by statistical data only driven models (Best et al., 2015; Haughton et al., 2016).





Given the uncertainties and/or limitations associated with LST estimates derived from satellites and LSMs, an alternative path to obtain increased accuracy LST estimates is to combine information from multiple sources. One limitation to this approach is the fact LST is not routinely measured at meteorological stations (e.g., Krishnan et al., 2015). Consequently, only a limited number of accurate in-situ LST estimates data records are available for calibration and evaluation, which do not provide the wide spatial and temporal coverage necessary for many applications. Nonetheless, previous works have shown potential added value may be obtained by combining information from satellite and LSMs (Ghilain et al., 2011; Martins et al., 2019). The most straightforward way to combine models and observations is to use satellite products to assess LSMs' LST estimates. Another example is the data assimilation of LST and other satellite-derived surface variables by numerical weather prediction models, which has been demonstrated to lead to improved forecasts accuracy of different atmospheric and surface variables at short to sub-seasonal scales (e.g., Schlosser and Dirmeyer, 2001; Pipunic et al., 2008; Ghent et al., 2010; Koster et al., 2010; de Rosnay et al., 2013; Bauer et al., 2015; Candy et al., 2017; Massari et al., 2018; Albergel et al., 2019; Sassi et al., 2019).

Several studies have pointed the benefit of remote sensing observations to constrain model surface parameters, which are otherwise difficult to measure directly. For example, previous works have employed visible, near infrared and microwave satellite measurements to derive surface roughness lengths over arid and semi-arid regions (Marticorena et al., 2004; Laurent et al., 2005; Prigent et al., 2012). Livneh and Lettenmaier (2012) used satellite-based evapotranspiration derived from MODerate-resolution Imaging Spectroradiomater (MODIS) and Geostationary Operation Environmental Satellites (GOES) together with in-situ and reanalysis datasets to constrain several surface hydrology parameters in a land surface mode. Orth et al. (2017) used several observational datasets (including satellite-based, in-situ, and combined model-observation products) of LST, evapotranspiration, soil moisture and terrestrial, water storage to constrain several parameters in the European Centre for Medium Range Weather Forecasts (ECMWF) Carbon-Hydrology Tiled ECMWF Scheme for Surface Exchanges over Land (CHTESSEL, van den Hurk et al., 2000; Balsamo et al., 2009; Dutra et al. 2010; Boussetta et al., 2013). They found that model simulations of LST are highly sensitive to the skin conductivity, the minimum stomatal resistance and the soil moisture stress function. Trigo et al. (2015) used the satellite-based LST estimates from the Satellite Application Facility Land Surface Analysis (LSA-SAF) dataset to constrain Leaf Area Index (LAI) and the surface roughness lengths for heat and momentum over most of Africa and Europe in the ECMWF Integrated Forecast System (IFS), which uses CHTESSEL as LSM. The changes to LAI were found to have a positive but limited impact on simulated LST, while the revised surface roughness lengths lead to overall positive impacts on LST. The improvements were particularly pronounced over arid and semi-arid regions (including wide portions of Iberia) where daytime skin temperature was affected by a significant cold bias in the original formulation.

Recently, Johanssen et al. (2019) (henceforth JO19) used the LSA-SAF dataset as reference to compare the estimates of summer daily maximum LST over Iberia between the ECMWF 4[th] and 5[th] generation reanalysis, respectively ERA-Interim and ERA5. ERA5 presented an overall improvement compared to ERA-Interim. However, both reanalysis displayed a large cold bias in the daily maximum LST over most of Iberia. Additionally, this cold bias was shown to be tightly related to the differences in the fraction of vegetation coverage (FCOVER) in the CHTESSEL land-surface model (which is the land surface component ERA5) and the satellite-based FCOVER estimates from the Copernicus Global Land Service (CGLS). By focusing on four grid-points in southern Portugal, JO19 found that a significant reduction of the LST cold bias could be obtained by correcting the model high and low vegetation fractions using the satellite-based European Space Agency Climate Change Initiative (ESA-CCI) land cover dataset. Finally, a sensitivity test showed that the model vegetation density parameter (which controls the percentage of bare soil in CHTESSEL) played a critical role for the summer daily maximum LST in these four grid-points. Similarly, Guo et al. (2019) used Leaf Area Index (LAI) estimates from MODIS to revise several vegetation coverage parameters over China in a regional climate model. The revised parameters were shown to have a positive impact on near-surface air temperature (a variable highly correlated with LST) and precipitation when compared to a gridded dataset interpolated from (station) in-situ measurements. Zheng et al. (2012) evaluated the LST from the National Centers for Environmental Prediction (NCEP) Global Forecast System (GFS) over the continental United Sates (CONUS) against GOES and in situ data for the summer of 2007. They found a large cold bias over the arid western CONUS during daytime, which



was largely reduced by including a new vegetation-dependent formulation of momentum and thermal roughness lengths in GFS.

Building on these previous works, here we use the LSA-SAF LST estimates, together with the CGLS FCOVER and LAI products and with the ESA-CCI vegetation cover dataset, to make an extensive revision of the vegetation cover in CHTESSEL over Iberia. This revision includes the model vegetation types and fractions, LAI and parameterization of vegetation density, which in turn impacts several other parameters such as, the surface roughness lengths for momentum and heat, skin conductivity. Additionally, we also evaluate the most recent version (v8.1) of Météo-France modelling platform SURFEX (SURface EXternalisée, Masson et al., 2013) which provides an additional constraint,

given its use of the recent ECOCLIMAP-II land cover database. The present manuscript is organized as follows: the datasets and models are described in Section 2; the vegetation coverage revision and the offline LSM simulations performed in the study are described in Section 3; the analysis of the ERA5 and control offline CHTESSEL simulation (with the original vegetation coverage) are described in Section 4; the results of the simulations with revised vegetation coverage are presented in Section 5; the results are discussed in Section 6, and the main conclusions of this study are

presented in Section 7.

## 2.    Data and Methods

### 2.1. Datasets

ERA5 is the latest global atmospheric reanalysis produced by the ECMWF, currently extending from 1979 to present (see Hersbach et al., 2018 for a detailed description of ERA5). It is based on a recent version of the ECMWF Integrated Forecast System (IFS cycle 41r2, more information at https://www.ecmwf.int/en/forecasts/documentation-and-support/changes-ecmwf-model/ifs-documentation, last access: November 2019), including several improvements compared to the version used in ERA-Interim (the ECMWF's previous generation reanalysis, Dee et al., 2011). Namely, ERA5 features increased temporal, horizontal and vertical resolutions (respectively 1 hour, ~31 km and 137

vertical levels extending from surface to 0.01 hPa). ERA5 also benefits from improvements to several model parameterizations (e.g. convection and microphysics) and to the four-dimensional variational data assimilation scheme (Hersbach et al., 2018), resulting in an overall better accuracy in representing several climate variables compared to ERA-Interim, including LST, near-surface air temperature, wind, radiation and rainfall (e.g. Urraca, 2018; Beck et al., 2019; JO19; Rivas and Stoffelen, 2019; Nogueira, 2020). Additionally, an increased number and more recent versions

of a wide variety of observational datasets are assimilated in ERA5. In this study, we use ERA5 reanalysis LST, total cloud cover (TCC) and surface evaporation fields, all retrieved at hourly frequency over a 0.25º × 0.25º regular grid for the 2004 to 2015 period from the Copernicus Climate Change Service Information website (https://climate.copernicus.eu/climate-reanalysis, last access: November 2019).

The LSA-SAF LST is derived from measurements performed by the Spinning Enhanced Visible and InfraRed Imager (SEVIRI) onboard the Meteosat Second Generation (MSG) series of satellites by employing the generalized "split-window" technique described in Freitas et al., (2010). The LSA-SAF LST estimates are available every 15 minutes for all the land pixels of the MSG disk, comprising satellite zenith view angles between 0º and 80º, with a resolution of 3 km at the nadir.

Estimates of daily surface evaporation were obtained from the Global Land Evaporation Amsterdam Model version 3.3b (GLEAMv3b, Martens et al., 2017). GLEAMv3b provides daily estimates of surface evaporative flux at 0.25ºx0.25º resolution covering the entire globe. It is based on the Priestley-Taylor expression (Priestley and Taylor, 1972) using only remotely sensed data.

The CGLS FCOVER and LAI estimates were obtained at 1km resolution covering the entire globe. Both CGLS LAI and CGLS FCOVER estimates are derived from the SPOT/VEGETATION and PROBA-V satellite observations using

the algorithm described by Verger et al. (2014).





The ESA-CCI land cover dataset was also considered in the present investigation. This dataset is derived by combining remotely sensed surface reflectance and ground-truth observations (Defourny et al., 2014). It provides consistent maps at 300 m spatial resolution on an annual basis from 1992 to 2015. It includes a total of 22 level-1 land cover classes and level-2 sub-classes, based on the land cover classification system developed by the United Nations Food and Agriculture Organization. In this study the 2010 data was chosen, considered to be representative of the full 1992-2015 period, since land cover changes over Iberia were very minor during that time interval.

## 2.2. Land surface models

CHTESSEL is the land-surface model of the ECMWF IFS which underlies ERA5. CHTESSEL represents a surface skin layer with zero heat capacity, which separates the atmosphere from the subsoil, and where the exchanges between surface and atmosphere take place. Each grid point of the skin layer can be divided into different tiles, representing different types of land cover: the dominant type of low and high vegetation, bare ground, intercepted water (on the canopy), and shaded and exposed snow. This information is then used to generate spatial fields of the surface parameters that control the land-atmosphere interactions (e.g., surface albedo, emissivity, momentum and heat roughness lengths, etc.).

The vegetation coverage in CHTESSEL corresponds to 2-dimensional static input fields. These fields provide, for each grid point the fraction of low vegetation (CVl), the fraction of high vegetation (CVh), the dominant type of low vegetation (TVl), and the dominant type of high vegetation (TVh). Additionally, the vegetation density parameters for low and high vegetation (respectively cvegl and cvegh, which range between 0 and 1) determine the fraction of the high and low vegetation tiles (Cl and Ch, respectively) that are effectively covered by vegetation, such that:

$$Cl = CVl \times cvegl$$

(1)

$$Ch = CVh \times cvegh$$

The effective grid cell total vegetation coverage is equal to $Cl + Ch$ and, thus, in neglecting snow and water bodies (which cover only a minor fraction of Iberia) the bare ground fraction is given by:

$$Cb = 1 - (Cl + Ch) \qquad (2)$$

In CHTESSEL, the vegetation fractions and types are derived from the Global Land Cover Characteristics (GLCC) data (Loveland et al., 2000). The vegetation density parameters for low and high vegetation are obtained from lookup tables as a function of the respective dominant type of vegetation (see Supplementary Table S1). Here the LST was derived from CHTESSEL as the temperature of the skin layer over each grid-box. This skin temperature is computed by the model from the surface energy balance equation calculated independently for each tile. The grid-box skin temperature is then defined as the weighted average of the LST on each tile fraction. This variable, defining the model's long-wave radiation emitted by the surface, is close to LST obtained from TIR observations, which in turn correspond to a radiative temperature of the surface within the satellite field of view (e.g., Trigo et al., 2015).

SURFEX (v8.1) was employed here using a CO2-responsive version of the Interaction between Soil Biosphere Atmosphere (ISBA) land-surface scheme, which includes interactive vegetation (Calvet et al., 1998; Gibelin et al., 2006). The ISBA 12-layer explicit snow scheme (Boon and Etchevers, 2001; Decharme et al., 2016) and its multilayer soil diffusion scheme (ISBA-Dif), with 14 layers and the 'NIT' biomass option were used. The ISBA parameters are defined for 12 generic land surface patches, including bare soil, rocks, permanent snow and ice, and nine functional



types (needle leaf trees, evergreen broadleaf trees, deciduous broadleaf trees, C3 crops, C4 crops, C4 irrigated crops, herbaceous, tropical herbaceous and wetlands). The land cover parameters (including vegetation coverage) in SURFEX are derived from the ECOCLIMAP-II database (Faroux et al., 2013), which is developed at a resolution of 1 km over Europe. ECOCLIMAP-II provides a static (representative of year 2000) land cover classification and associated surface parameters based on Corine Land Cover map over Europe. It also uses other auxiliary data sources

to derive the land cover classification and parameters, including the leaf area index (LAI) from MODIS (MODerate resolution Imaging Spectroradiometer) and the normalized difference vegetation index (NDVI) from SPOT/Vegetation satellite mission.

### 3. Land-surface simulations and revised vegetation coverage

**3.1. Simulation description**

An offline land-surface simulation covering Iberia was generated using CHTESSEL forced by ERA5 fields - namely near-surface (10m) air temperature, humidity and wind speed, surface pressure, rainfall and solar and thermal downwelling radiative fluxes. The simulation domain corresponded to a regular grid covering the region from 35ºN to 45ºN and from 5ºW to 10ºW at 0.25° × 0.25° resolution (see Figure 1). The run was initialized in 2002 and extended

until the end of 2015, with a 15-minute time-step. However, it should be noticed that the analysis of the simulation results is performed over the 2004-2015 period, with the first two years discarded due to model spin-up. This simulation represents a baseline for the CHTESSEL simulations with updated vegetation coverage presented below and, thus, was named CTR (control). A similar land-surface offline simulation was generated with SURFEX, henceforth denoted SFX, which covered the same spatial domain and period and was forced by the same ERA5 fields.

In addition to CTR and SFX, three additional simulations were performed with CHTESSEL, representing three different levels of updated vegetation coverage compared to CTR (see Table 1 for a summarized description of the main characteristics of the simulations considered in the present investigation). These additional simulations were carried over the same domain and period.

The first revised simulation, denoted H_CCI, focused on the vegetation fractions and types. In this revision, the high

and low vegetation fractions and types were replaced in all grid-points over Iberia by the data derived from the ESA-CCI Land cover dataset. The original 300 m ESA-CCI land cover classes were spatially aggregated to the 0.25°×0.25° regular grid, generating fields with the fraction of each of the ESA-CCI 22 land cover class. These fractions were computed by counting the number of 300 m pixels of each class occurring within each of the 0.25°×0.25° grid-cells. These fractional classes were then converted to the CHTESSEL land cover classes using a cross-walking table adapted

from Poulter et al. (2014). These new land cover fractions are then processed to compute the fractional cover of low and high vegetation (CVl, CVh) and the dominant types of low and high vegetation attributed to each grid-point. The vegetation density parameters (and other parameters associated with each dominant vegetation type) were obtained from the CHTESSEL lookup tables (see table S1), analogous to the procedure in CTR, but taking the update vegetation types and cover into account. The maps of CVl and CVh of CTR and H_CCI are shown in Supplementary Figure S1,

along with a comparison between the dominant low and high vegetation types in CTR and H_CCI.

The two other simulations, denoted H_CCI_cl and H_CCI_cl_LAI, used the same updated vegetation based on ESA-CCI as H_CCI, but the low and high vegetation density parameters were revised. Following Alessandri et al. (2017), we employed clumping for high and low vegetation in both simulations, by introducing a Lambert-Beer law exponential dependence on LAI for both cvegl and cvegh:

$$cvegl = 1 - exp(-k_1 \times LAI) \tag{3}$$



$$cvegh = 1 - exp(-k_{\mathrm{h}} \times LAI)$$

Equation 3 introduces a representation of the seasonal cycle for the high and low vegetation coverage. Notice that the vegetation clumping employed here is similar to the computation of cvegl in SURFEX (Le Moigne, 2018). However, in SURFEX cvegh is obtained from lookup tables based on vegetation cover types. In a preliminary analysis we tested the clumping for high and low vegetation or only for low vegetation in CHTESSEL. The results showed better performance of LST simulation when clumping was introduced for both high and low vegetation. Clumping only for
low vegetation required further changes to the surface roughness parameter for high vegetation as discussed in Section 4 below. Here a value of $k_h=k_l=0.6$ was considered for equation 3, which is the same value used in SURFEX for cvegl. Alessandri et al. (2017) used $k_h=k_l=0.5$, but the impact of using 0.5 or 0.6 in the LST simulation was reduced.

The difference between H_CCI_cl and H_CCI_cl_LAI simulations was in the LAI fields provided as input to CHTESSEL. Notice that in all CHTESSEL simulations, LAI is prescribed as a mean monthly climatology for each
grid-point. Simulation H_CCI_cl used the original LAI fields in CHTESSEL (as in CTR and H_CCI). But H_CCI_cl_LAI used the LAI fields from the CGLS database. For this purpose, the 10-daily version 2 CGLS LAI was processed for the period 1999-2018 to generate the mean monthly climatology computing the monthly mean for each calendar month in the full 20-years period. This generated a climatology at 1 km that was then aggregated to the regular 0.25°x0.25° resolution by mapping the 1 km grid into the coarser grid. The motivation for updating the LAI
fields in H_CCI_cl_LAI is provided by the high sensitivity of maximum daily LST on cveg reported by JO19. Consequently, equation 3 should introduce a significant sensitivity of daily maximum LST on LAI.

### 3.2. Simulation evaluation metrics

An overall evaluation of the skin temperature estimates derived from the considered model-based datasets is shown in Table 2, using the LSA-SAF LST product as reference. Since LSA-SAF LST is a clear-sky variable, the comparison
against model skin temperature (or model LST) must be performed for model clear-sky cases as well. Restricting the analysis to summer months (June-July-August, JJA) ensures a reasonable sample of both model and satellite LST values, since cloud coverage remains relatively low in Iberia during summer months, as shown by JO19. Additionally, for the comparison between simulated and observed LST to be consistent, we performed an upscaling of the LSA-SAF LST hourly data, by computing the median of the whole group of LSA-SAF LST pixels (at 3 km resolution)
inside each 0.25°x0.25° grid cell. The fraction of valid pixels (each grid cell and time) was retained during the upscaling procedure and used as a proxy to compute clear sky fraction. Subsequently, for each 0.25°x0.25° grid cell over Iberia, only the instants where the percentage of valid LSA-SAF pixels was greater than 70% and the ERA5 total cloud cover was below 30% were considered for dataset comparison. These two thresholds were chosen based on JO19, corresponding to a balance between ensuring most of the grid cell to be cloud-free while keeping a large amount
of valid data for the JJA 2004-2015 period over Iberia.

The daily maximum LST (LSTmax) was computed as the maximum value over the 11 to 18 UTC interval, while daily minimum LST (LSTmin) was computed as the minimum over the 00 to 07 UTC interval. These ranges were chosen to avoid the identification of daily extremes during time periods which are not representative of the minimum nocturnal and maximum afternoon temperature (for example, under cloudy daytime and clear-sky night-time conditions, one
would identify LSTmax during the latter).

Although the sample of LSA-SAF LST estimates was reduced outside the summer months, we compared the seasonal cycle of daytime maximum and night-time minimum LST amongst the different model-based datasets considered in the present investigation. This comparison provides a quantitative measure of the impact of the use of different land-surface models and different vegetation coverage formulations on all seasons.

Additionally, we estimated the maximum monthly FCOVER over Iberia in the different simulations considered here and compared against the CGLS FCOVER. Here the assumption was that maximum monthly FCOVER derived from





satellite corresponds to 1 minus the permanent bare soil coverage. In turn, the percentage of bare soil may be estimated from land-surface models from by employing equation 2. Notice that in other months (where FCOVER is not maximum), the dependence of the fraction of green vegetation coverage on the LAI, which is not represented in the original CHTESSEL formulation, may render this approximation invalid. We also compared the fractions of low and high vegetation (i.e. Cl and Ch, respectively) amongst the different simulations. When analysing these comparisons amongst the vegetation coverage in the different simulations, it is important to keep in mind that the CGLS FCOVER is tightly related to the CGLS LAI (they are derived from the same observations) which is given as input for H_CCI_cl_LAI. Additionally, the ESA-CCI vegetation cover is also used in constructing the ECOCLIMAP-II used by SURFEX. Nonetheless, these comparisons provide relevant insights into the differences in the LST fields amongst the different simulations as discussed below.

Finally, we also compared the surface evaporation from the different simulations considered here and from the GLEAMv3b dataset over the different seasons. Surface evaporation estimates from GLEAMv3b have non-negligible uncertainties and should be regarded as an additional product rather than the truth. Nonetheless, this comparison provides a quantitative measure into the potential impact of the changes in vegetation cover and LST to the surface water and energy balances over Iberia.

## 4. Results

### 4.1. Evaluation of summer daily maximum LST over Iberia

The daily maximum LST from ERA5 displayed a large cold bias over Iberia during JJA months (JO19 and Fig. 1a), with magnitudes larger than 10ºC over wide portions of central and southwestern Iberia. Fig. 1b shows that the spatial pattern of daily maximum LST bias over Iberia in ERA5 was very closely reproduced by the control CHTESSEL offline simulation (CTR). This result opened a path to investigate, attempt to understand and correct the LST error sources over Iberia focusing on limited area CHTESSEL offline simulations, rather than running the full global coupled IFS model.

The spatial pattern of JJA daily maximum LST bias over Iberia simulated by SURFEX (SFX, Fig. 1c) was clearly distinct from CTR. Overall SFX displayed a smaller magnitude and positive bias over most Iberia, except for the northernmost regions. The lower magnitude of the JJA LSTmax over Iberia in SFX compared to CTR and ERA5 was further evidenced by comparing the JJA daily max LST RMSE maps (Fig. 2a-c). The RMSE averaged over all Iberia grid-points was 3.2ºC in SFX, well below the 5.7ºC value found for ERA5 and CTR. Similarly, the JJA LSTmax bias averaged over all Iberia grid-points was 1.1ºC in SFX, but -5.1ºC in ERA5 and -5.0ºC in CTR (see Table 2 for a summary of the overall errors in all the considered simulations). The large differences in the LSTmax errors between ERA5/CTR and those of SFX suggest that the errors in CTR are not due to the atmospheric forcing since SFX was driven by the same data. Moreover, the spatial patterns of RMSE were also nearly identical between ERA5 and CTR providing further support to the use of offline CHTESSEL simulations to investigate the causes of the LST errors in ERA5. Additionally, these spatial patterns were tightly related to the corresponding bias patterns in Fig. 1a-c, highlighting the relevance of the systematic bias to the overall error in the summer LSTmax over Iberia in the ECMWF products.

### 4.2. Relation between the Iberia LST bias and the vegetation cover

Fig. 3 shows scatter plots of JJA LSTmax bias versus the maximum monthly FCOVER error over Iberia. The latter is estimated from absolute differences between model and CGLS fractions of green vegetation cover. The results show a relation between the LSTmax bias and the absolute error in maximum FCOVER in ERA5 (Fig. 3a) and CTR (Fig. 3b), with correlation coefficients of -0.4 and -0.5 respectively, and with largest LSTmax bias occurring for large FCOVER absolute errors. Furthermore, we found that, in general, the maximum monthly FCOVER in ERA5 and CTR (Fig. 4b) largely overestimated the CGLS FCOVER (Fig. 4a) over most of the regions of Iberia where large daily





maximum LST bias (cf. Fig. 1) and RMSE (cf. Fig. 2) value were found for these datasets. We point out that the vegetation cover fields are identical in ERA5 and CTR. In contrast, SFX displayed lower values for both LST bias and absolute maximum monthly FCOVER error, and with a correlation coefficient of only 0.2 between these two

errors (Fig. 3d). Furthermore, the maximum FCOVER pattern estimated from SFX (Fig. 4c) was closer to the CGLS dataset, although SFX also tends to overestimate the FCOVER. Given these results, it is hypothesized that, at least part of the error in daily maximum LST during summer over Iberia by CHTESSEL (and thus in ECMWF products) was due to its misrepresentation of the vegetation coverage in this region. This hypothesis was also recently put forward in JO19, which reported a similar relation between the summer LST bias and the misrepresentation of summer

FCOVER over Iberia. Here, this hypothesis was further supported by the significantly improved representation of the vegetation coverage over Iberia in SURFEX (largely due to its use of the ECOCLIMAP-II database), corresponding to a drastic reduction of the summer LST errors over Iberia.

Subsequently, we compared the fractions of high and low vegetation (i.e. Ch and Cl) between the original CHTESSEL based products (ERA5 and CTR) and SFX during JJA. On the one hand, the original CHTESSEL based products

overestimated Cl over wide portions of Iberia, particularly in the North (Fig. 5a) compared to SURFEX (Fig. 5c). On the other hand, ERA5 and CTR displayed a significantly larger Ch values (Fig. 5b) compared to SFX (Fig. 5d) throughout most of Iberia, except for the north-western region where the reverse is true. We notice that many of the regions with largest overestimation of Ch in CHTESSEL when compared to SURFEX, correspond to regions with the largest cold bias in summer daily maximum LST (cf. Fig. 1) and largest magnitude of RMSE (cf. Fig. 2) in ERA5 and

CTR. The latter was well illustrated by mapping the difference in RMSE between SFX and CTR (Supplementary Figure S2a). Thus, we propose that the misrepresentation of the high and low vegetation coverages in CHTESSEL is a key source for the systematic cold bias found for the simulated summer daily maximum LST over Iberia.

### 5. Correcting the vegetation coverage over Iberia on CHTESSEL

#### 5.1. Impact of updated vegetation on JJA daily maximum LST

The results presented in Section 4 highlight the large cold bias affecting summer daily maximum LST over Iberia in the ECMWF products. In contrast, given the same atmospheric forcing, SURFEX displayed a lower magnitude, and positive, LST bias. One of the key differences between the two models was the representation of vegetation coverage over Iberia. In this section, we evaluate the impact of the changes in vegetation coverage in the LST simulations.

First, in simulation H_CCI the high and low vegetation fractions and types were modified based on ESA-CCI database. Compared to CTR, H_CCI did not show a consistent reduction of summer daily maximum LST bias (Fig. 1d) nor RMSE (Fig. 2d) throughout most of Iberia. This is also illustrated by the bias and RMSE boxplots computed from CTR and H_CCI in Fig. 6. Although the largest magnitude errors and overall error spread are reduced in H_CCI, the median increased slightly compared to CTR. Averaged over Iberia, the overall bias increased from -5.0ºC to -5.8ºC

and the overall RMSE increased from 5.7ºC to 6.2ºC (see Table 2). The reason for the poor performance of H_CCI may be understood by analysing its vegetation cover. Although H_CCI displayed a reduced correlation between LST bias and maximum monthly FCOVER absolute error (r=-0.1, Fig. 3d), it also displayed an overall larger overestimation of the maximum monthly FCOVER (Fig. 4d) when compared to CTR (Fig. 4b). The cause for this increased FCOVER was the widespread overestimation of the fraction of low vegetation (Figs. 5e), despite the fraction

of high vegetation being significantly closer to SFX (Fig. 5f) (ECOCLIMAP-II is assumed here to have a more realistic vegetation coverage than the original CHTESSEL formulation).

The large overestimation of the fraction of low vegetation in H_CCI was due to an overestimation of the low vegetation density parameter, cvegl. This is a critical parameter in determining the amplitude of the diurnal cycle of LST over Iberia during summer months, as shown by JO19. In the original CHTESSEL formulation, the vegetation density

parameters were obtained from lookup tables, based on the dominant vegetation types for each grid-box. With the clumping parameterization introduced in H_CCI_cl, the vegetation density parameters were estimated as a function



of the LAI, following equation 3, in this case the default LAI data used in the CTR simulation was used. This resulted in a moderate reduction of the summer fraction of low vegetation (Cl, Fig. 5g) compared to CTR and to H_CCI, while the fraction of high vegetation remained nearly identical to H_CCI (Ch, Fig. 5h). Consequently, the Cl reduction lead to a moderate reduction of the maximum monthly FCOVER in H_CCI_cl (Fig. 4e) compared to CTR (Fig. 4b) and to H_CCI (Fig. 4d). As expected, the increased amount of bare soil (by reducing the vegetation coverage) leads to enhanced daytime warming, thus reducing the magnitude of the cold LSTmax bias over most of Iberia (Fig. 1f) and, consequently, also reducing the RMSE (Fig. 2f). The overall summer LSTmax bias averaged over entire Iberia was -4.4°C in H_CCI_cl, -5.0°C in CTR, and -5.8°C in H_CCI. The overall RMSE reduced from 5.7°C in CTR (and 6.2°C in H_CCI) to 5.0°C in H_CCI_cl. The bias and RMSE boxplots in Fig. 6 also illustrate this reduction, although they also evidence that the errors in summer LSTmax in SFX were lower than in H_CCI_cl.

The maximum monthly FCOVER from H_CCI_cl (Fig. 4e) still represents an overestimation of the CGLS dataset (Fig. 4a), although slightly improved compared to CTR (Fig. 4b) and H_CCI (Fig. 4d). The pattern of summer fraction of high vegetation in H_CCI_cl (Fig. 5h) was similar to H_CCI (Fig. f) and SFX (Fig. 5d). However, the large overestimation of the summer fraction of low vegetation in H_CCI (Fig. 5e) was only moderately decreased in H_CCI_cl (Fig. 5g), and still larger than the values found in SFX over most of Iberia (Fig. 5c).

Updating the vegetation fractions and types based on ESA-CCI resulted in major changes in the areas dominated by high and low vegetation between CTR and H_CCI or H_CCI_cl (Fig. 4). However, the LAI field used in H_CCI_cl to compute the vegetation density parameters was not updated, and corresponds to the original vegetation distributions in CTR. We argue that this can generate some inconsistencies and limits the potential improvement in the representation of summer LST in H_CCI_cl. Consequently, a third simulation was introduced (denoted H_CCI_cl_LAI), which includes the revised vegetation fractions and types based on ESA-CCI, the clumping parameterization and updated LAI fields from CGLS database. Fig. 5 shows that the summer fractions of high vegetation in H_CCI_cl_LAI are also similar to CTR, H_CCI and H_CCI_cl. However, it also shows that the summer fraction of low vegetation was significantly reduced in H_CCI_cl_LAI compared to the other CHTESSEL simulations, rendering its pattern much closer to SFX.

The maximum monthly FCOVER was also reduced in H_CCI_cl_LAI over wide portions of Iberia, resulting in a closer match to the CGLS FCOVER compared to the previous CHTESSEL simulations (Fig. 4). This match in FCOVER between H_CCI_cl_LAI and CGLS was not surprising since the CGLS LAI and the CGLS FCOVER are tightly related to each other. A better, more independent way, to evaluate the impact of the revised vegetation cover is to analyse the simulated summer LSTmax fields. The results showed that the overall bias averaged over entire Iberia was -1.5°C for H_CCI_cl_LAI. This was much lower than the values of -4.4°C in H_CCI_cl and -5.0°C for CTR, and closer in magnitude to the 1.1°C found for SFX. Similarly, the overall RMSE was 3.5°C for H_CCI_cl_LAI, 5.0°C in H_CCI_cl, 3.2°C in SFX and 5.7°C in CTR. The boxplots of summer daily maximum LST bias and RMSE over Iberia confirm the large error reduction in H_CCI_cl_LAI compared to all other CHTESSEL simulations, with the overall performance becoming very close to SFX. This is also illustrated by comparing the spatial patterns of bias and RMSE in Fig. 1 and Fig. 2 respectively. Finally, it should be noted that LST biases and RMSE's attained by SFX and H_CCI_cl_LAI are already very close to the accuracy values generally attributed to satellite LST values (Trigo et al, 2011; Göttsche et al., 2016).

We compared the full diurnal cycle of the summer LST in all datasets considered here, in order to assess the differences in the Iberia summer LST amongst datasets outside of the time of daytime maximum. Fig. 7 shows the JJA LST at each hour of the day averaged over the 2004-2015 period and over all Iberia grid-points (only considering clear-sky conditions in both satellite and simulations). The average warming rate during the morning in LSA-SAF was overestimated in SFX, resulting in a slight warm bias of LSTmax, but underestimated by CHTESSEL, resulting in the cold bias found for CTR and ERA5. H_CCI displayed a slightly larger underestimation of the morning warming rate compared to CTR, thus increasing the magnitude of the cold bias. H_CCI_cl showed only slight increase of the morning warming rate compared to CTR, but still well below the LSA-SAF warming rate. Finally, in H_CCI_cl_LAI the morning warming rate was much closer to LSA-SAF, resulting in larger reduction of the LSTmax cold bias, of similar magnitude to the warm bias in SFX. One must notice the slight shift of the time of maximum LST in all



simulations using CHTESSEL (ERA5, CTR, H_CCI, H_CCI_cl and H_CCI_cl_LAI) to 14 UTC, one hour later than for LSA-SAF. We also notice that SFX did not display phase shift, peaking at 13 UTC. The time delay to reach maximum temperature in all CHTESSEL simulations is likely related with a too strong coupling with the underlying soil, but further investigation of this issue is beyond the scope of this study. During the afternoon the cooling rate tends to be slightly slower in all models than in observations, but the resulting differences in night-time LST are

relatively small amongst all datasets (below 2ºC), and within the typical uncertainty associated with the LSA-SAF LST estimates (Trigo et al., 2011).

## 5.2. Impact of updated vegetation on LST outside summer

        The seasonal cycle of LSTmax and LSTmin averaged over Iberia during the 2004-2015 period amongst all model-

based datasets is shown in Fig 8a. Notice that outside JJA, the comparison with LSA-SAF LST was not performed due to frequent cloud cover over Iberia, reducing the satellite sample and increasing the probability of cloud contamination, as discussed above. However, one may still assess the impact of the different model formulations in simulating the Iberia LST outside of summer. This is particularly relevant for simulations employing clumping (H_CCI_cl, H_CCI_cl_LAI and SFX), since equation 3 introduces a seasonal cycle in the vegetation density

parameters.

        The monthly mean LSTmax averaged over Iberia was identical between ERA5 and CTR throughout the entire year (Fig. 8a), providing further robustness to the use of the offline simulations to assess the errors of LST in ERA5 dataset and their main sources. Fig. 8a also shows that the differences in monthly averaged daily maximum LST between H_CCI and CTR and between H_CCI_cl and CTR were below 1ºC throughout the entire year. Large differences

emerged between H_CCI_cl_LAI and CTR during the summer months, up to 3.8ºC. It should be noticed that this comparison differs from the diurnal cycle results in Figure 7, since it represents all weather conditions while the diurnal cycle in Figure 7 represents clear-sky only conditions in both LSA-SAF and ERA5. However, these differences reduced significantly over the colder months, being below 1.0ºC between November and April. A similar result was found when comparing the average seasonal cycle daily maximum LST between SFX and CTR, with SFX being

warmer up to 5.9ºC during summer, but only ~0.5ºC during winter months.

        The seasonal cycle of LSTmin averaged over Iberia (Fig. 8b) revealed that all the considered offline simulations (CTR, H_CCI, H_CCI_cl, H_CCI_cl_LAI and SFX) underestimated this variable compared to ERA5 throughout the entire year, with the largest underestimation occurring during summer. However, the magnitude of this summer underestimation was below 1.2ºC in all cases. Additionally, the differences in monthly mean LSTmin averaged over

Iberia were below 0.25ºC between CTR and simulations H_CCI, H_CCI_cl and H_CCI_cl_LAI. These results show that the updated vegetation coverage had little impact on the night-time minimum temperature throughout the entire year, in agreement with the results of the JJA LSTmin diurnal cycle in Fig. 7.

## 6. Discussion

Simulation H_CCI_cl_LAI features an updated representation of vegetation over Iberia compared to the original CHTESSEL formulation, namely with revised LAI and vegetation fractions and types, and a clumping parameterization for the vegetation density parameters. Compared to the CTR simulation, H_CCI_cl_LAI resulted in a closer agreement to the fractions of low and high vegetation during summer in SFX, derived from ECOCLIMAP-II. Additionally, it also resulted in a closer match to the maximum monthly FCOVER from the CGLS dataset. Both

these comparisons must be interpreted with care. On the one hand, H_CCI_cl_LAI uses the CGLS LAI, which is turn is tightly related to the CGLS FCOVER. On the other hand, ECOCLIMAP-II uses information on the vegetation cover from the ESA-CCI dataset. Furthermore, SFX also uses a clumping parameterization for the low vegetation density parameter (but not for the high vegetation parameter).





Nonetheless, a robust support to the added value of the corrections implemented in H_CCI_cl_LAI was provided by
comparing the simulated LST amongst datasets. Our results showed a large reduction of the summer daily maximum
LST cold bias in H_CCI_cl_LAI when compared to CTR. Additionally, we found relatively small differences (within
typical LST observation uncertainty; Göttsche et al, 2016) amongst these two datasets during winter and during night-
time. This suggests that the implemented changes are robust, in the sense that the simulated LST is improved during
daytime in warm months, with negligible impacts in all other months and hours of the day. Furthermore, the fact that
SFX displayed similar performance in the simulation of LST provides further support to the results of H_CCI_cl_LAI.

Interestingly, simulation H_CCI did not reduce the cold bias in summer daily maximum LST over Iberia, while
H_CCI_cl resulted only in a slight reduction. These results highlight an important point: the revision of the vegetation
coverage in the CHTESSEL model must be implemented consistently throughout its multiple dimensions. Updating
the vegetation types and fractions based on ESA-CCI from the original CHTESSEL formulation resulted in an error
reduction for the JJA LSTmax over the regions affected by the largest bias (Supplementary Figure S2b), which were
dominated by high vegetation in the original CHTESSEL and become dominated by low vegetation in H_CCI.
However, over the regions which were dominated by low vegetation in the original CHTESSEL formulation and
become dominated by high vegetation in H_CCI, the RMSE increased (cf. Supplementary Figure S2b and Fig. 5).
Sensitivity tests over these regions of increased error suggest that this was due to an overestimation of the low
vegetation density parameters. Indeed, introducing clumping in H_CCI_cl slightly reduced the cvegl parameter over
these regions, reducing the errors (Supplementary Figure S2c). However, the LAI used for the clumping
parameterization in H_CCI_cl corresponds to the original CHTESSEL data, where the grid-points dominated by low
and high vegetation were essentially reversed when compared to ESA-CCI (Fig. 5). Thus, updating the LAI becomes
necessary after updating the vegetation fractions and types and introducing clumping. Indeed, H_CCI_cl_LAI results
in significantly larger error reduction over the regions of large cold bias, and only very slight error increases (within
observation uncertainty) over the regions where the updated model is dominated by high vegetation (Supplementary
Figure S2d). Similarly, SFX which features similar vegetation types and fractions, clumping for low vegetation and
interactive LAI (thus coherent with its vegetation cover) resulted in much lower bias compared to CTR over regions
dominated by low vegetation, and similar bias over regions dominated by high vegetation (Supplementary Figure
S2a).

We point out that further improvement of the simulated surface temperature may be obtained by changing other
relevant surface parameters. For example, we tested changing the surface roughness length for heat transfer, z0h,
which plays an important role for the surface skin layer energy budget. We notice that in the original CHTESSEL
formulation, z0h is equal to surface roughness length for momentum, z0m, only for high vegetation dominated regions,
while for low vegetation dominated regions z0h is lower than z0m by a factor of 100 (see Supplementary Table S1).
Reducing the roughness length for heat transfer for high vegetation types in simulation H_CCI_cl_LAI by a factor of
10 results in a further reduction of the daily maximum LST errors over the regions dominated by high vegetation,
essentially removing all error increases compared to CTR (Supplementary Figure S2e). It is important to highlight
that, once again, the roughness length change is not effective by itself, but only when performed together with all other
corrections in H_CCI_cl_LAI. Indeed, Supplementary Figure S2f shows that reducing the roughness length for heat
transfer by a factor of 10 for high vegetation types directly in simulation CTR results in an overall increase of the
errors over most of Iberia (particularly over all regions that were dominated by high vegetation in CTR). Other surface
parameters also play an important role in simulated surface temperature, such as albedo and emissivity, are also related
to the surface vegetation coverage in LSMs (e.g. Nogueira and Soares, 2019).

Further support to the relevance a systematic and coherent update of all dimensions of the surface vegetation coverage
was provided by introducing the clumping parameterization directly in CTR, without updating LAI nor the vegetation
fractions and types. This resulted only in moderate error reduction throughout Iberia (Supplementary Figure S2g),
representing a much lower improvement when compared to H_CCI_cl_LAI.

Finally, the evaluation of other relevant surface variables is critical. For example, the changes to the vegetation cover
and LST between H_CCI_cl_LAI and CTR should have a relevant impact on the surface energy and water budgets.
Thus, evaluating the simulated surface water variables and water and energy fluxes is required for a robust validation





of the implemented changes. However, these fields are strongly dependent on the surface-atmospheric feedbacks and its validation based on offline simulation alone is not robust. Furthermore, there are very few reliable observational datasets over Iberia for these variables with appropriate accuracy to perform an effective model validation. In order to

illustrate this problem, we compared the surface evaporation amongst the simulation datasets and with GLEAMv3b (Fig. 8c). During August to February, ERA5 and all offline simulations displayed a similar magnitude overestimation of GLEAMv3b evaporation. Surprisingly, the best performing simulations in terms of summer LSTmax (i.e. SFX and H_CCI_cl_LAI) underestimated the March to July from GLEAMv3b by a larger degree than CTR. Interestingly, the increased underestimation occurs over north-western Iberia, which was dominated by low vegetation in CTR, but was

dominated by high vegetation in H_CCI_cl_LAI and SFX (Supplementary Figure S3). However, we notice that the seasonal cycle of Iberia average surface evaporation shows large differences between ERA5 and offline CHTESSEL, particularly between March and July, where ERA5 shows a large underestimation of evaporation (with the underestimation of peak evaporation larger than for H_CCI_LAI) when compared to GLEAMv3b (Fig. 7c). The differences between ERA5 and CTR (which are primarily due to the soil moisture land data assimilation increments

in ERA5) are of the same order of magnitude as the differences between CTR and H_CCI_cl_LAI. Thus, we conclude that an evaluation of the surface energy and water budget in coupled land-atmosphere simulation is required, which is the subject of a subsequent work.

## 7. Conclusions

We used the LSA-SAF satellite product to evaluate the summer LST over Iberia simulated by two LSMs - CHTESSEL and SURFEX – during the 2004-2015 period. Both LSMs were ran offline forced by the same atmospheric forcing fields obtained from the recently released ERA5 reanalysis.

The results show a large cold bias of the JJA LSTmax simulated by CHTESSEL, reaching magnitudes larger than 10ºC over wide portions of central and southwestern Iberia. This large cold bias was shown to be tightly linked to a

misrepresentation of the Iberia vegetation cover in CHTESSEL, when compared against ESA-CCI land cover dataset, CGLS FCOVER and ECOCLIMAP-II dataset. We show that this misrepresentation includes significant differences between the areas dominated by high and low vegetation types, but also the amount of bare soil and the fractions of green cover. This result agrees with recent investigations reporting a tight link between the misrepresentation of the vegetation coverage and the errors in LST simulated by different LSMs over different regions of the globe (Zheng et.,

2012; Trigo et al., 2015; Gu et al., 2019; and JO19). Accordingly, the summer LSTmax over Iberia simulated by SURFEX displayed a much smaller magnitude (positive) bias, which was within the observational uncertainty associated with LSA-SAF LST. One of the key reasons for this difference is the fact that SURFEX uses the updated land cover dataset ECOCLIMAP-II, and it includes interactive vegetation evolution, with a clumping parameterization for the low vegetation density parameter.

We propose a methodology to improve the representation of vegetation over Iberia in CHTESSEL by combining information from the ESA-CCI land cover dataset with the CGLS LAI. The proposed improvement in vegetation includes an update of the LAI data and high and low vegetation fractions and types, together with introducing a clumping parameterization for both high and low vegetation density. The clumping introduces seasonality to the vegetation coverage due to an exponential dependence on the LAI. We demonstrate that the proposed improved

representation of vegetation has significant added value, removing the daily maximum LST summer cold bias completely while never reducing the accuracy over all seasons and hours of the day.

By analysing different simulations with different degrees of changes to the vegetation coverage in CHTESSEL, we show that in order for the vegetation revision to be effective, it must be implemented consistently across its multiple dimensions. Specifically, we show that updating the vegetation types and fractions, without a coherent update to the

vegetation density parameters results in a slight degradation of the simulated LST compared to the original CHTESSEL simulation. Additionally, revising the vegetation density parameters by introducing the clumping parameterization introduces a strong dependency of the vegetation fraction on the LAI, which has only moderate



benefit if the LAI fields are not revised. It is the combination of these multiple coherent improvements that results in a large bias reduction, towards error magnitudes that are within values of the observational uncertainty. We point out that the proposed revised vegetation in CHTESSEL reduces the errors in JJA LSTmax without degrading the simulated LST for other times of the day or for other months of the year (to within the typical uncertainty associated with LSA-SAF estimates).

By performing some additional sensitivity tests, we showed the importance of the patterns of surface roughness lengths of momentum and heat transfer (z0m and z0h) for LST simulated by LSMs. Indeed, the satellite derived revision of the LAI and high and low vegetation types and fractions together with the clumping parameterization provides a path for an informed revision of the z0h and z0m patterns, which are very difficult to observe directly. Further improvements may be obtained by tuning the surface roughness parameters associated with the different types of vegetation or by improvements to the physical parameterization for soil heat diffusion and surface heat fluxes.

Finally, we point out that in the present study the proposed formulation was only tested for the simulated LST results. It is important to assess the results of the revised vegetation coverage in other components of the surface water and energy budgets. However, as shown here for evaporation, this requires coupled land-atmosphere simulations which will be the subject of a subsequent investigation. Furthermore, it requires accurate validation datasets which are not always available. Additionally, the proposed methodology must be tested over other domains outside Iberia.

Nonetheless, the present work has important implications. First, LST plays a central role in the surface-atmosphere energy and water exchanges and, thus, its accurate representation in LSMs is crucial for accurate Earth System Models. Second, CHTESSEL is the LSM employed by ECMWF in the production of their weather forecasts and reanalysis, hence systematic errors in CHTESSEL are expected to affect these products. Indeed, we show that pattern of the summer daily maximum LST cold bias over Iberia in CHTESSEL is also present in the widely used ERA5 reanalysis, while JO19 showed that it was also present in the ECMWF previous generation reanalysis ERA-Interim. In fact, the remarkable similarity in LSTmax error patterns between ERA5 and the offline CHTESSEL simulation provides a robust basis for using the offline setup employed here to assess and improve the simulation of LST. Finally, our results provide hints into the role played by vegetation in land-atmosphere exchanges, highlighting the relevance of consistent vegetation cover and corresponding seasonality for LST simulated by both models, as well as pointing out how Earth observations may be used for constraining and improving weather and climate simulations.

## Acknowledgements

This research was funded by Fundação para a Ciência e a Tecnologia (FCT) grant number PTDC/CTA-MET/28946/2017 (CONTROL). E. Dutra was funded by FCT research grant IF/00817/2015. The authors would also like to acknowledge the financial support of FCT through project UIDB/50019/2020 – IDL.

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

**Code Availability**

The SURFEX modelling platform of Meteo-France is open-source and can be downloaded freely at http://www.umr-cnrm.fr/surfex/ (CNRM, 2016) and uses the CECILL-C licence (a French equivalent to the L-GPL licence; http://cecill.info/licences/Licence_CeCILL_V1.1-US.html; CEA-CNRS-Inria, 2013). It is updated at a relatively low frequency (every 3 to 6 months). If more frequent updates are needed, or if what is required is not in
Open-SURFEX (DrHOOK, FA/LFI formats or GAUSSIAN grid), you are invited to follow the procedure to get an SVN account and to access real-time modifications of the code (see the instructions in the first link). The developments presented in this study stem from SURFEX version 8.1.

The ECMWF land surface model configurations described here are based on CHTESSEL model. The CHTESSEL source code is available subject to a license agreement with ECMWF. ECMWF member state weather services and
their approved partners will be granted access. The CHTESSEL code without modules related to data assimilation is also available for educational and academic purposes as part of the OpenIFS project (https://software.ecmwf.int/wiki/display/OIFS/OpenIFS+Home, and https://confluence.ecmwf.int/display/OIFS/Offline+Surface+Model+User+Guide). Further details regarding data availability and model configurations, including the information required to reproduce the presented results on
ECMWF systems, are available from the authors on request.

     **Data/Code Availability**

ERA5 data can be obtained freely from the Copernicus Climate Change Service Information website (https://climate.copernicus.eu/).

The CGLS FCOVER and LAI can be obtained freely from the CGLS website (https://land.copernicus.eu/)

The ESA-CCI land cover can be obtained freely from their website (https://www.esa-landcover-cci.org/)

The LSA-SAF LST can be obtained freely from their website (https://landsaf.ipma.pt/).



All the considered fields (LST, evaporation, LAI, FCOVER, Ch and Cl) from all the considered simulations considered in the present study as well as the source code of SURFEX-ISBA model are freely available

at http://doi.org/10.5281/zenodo.3701230. Due to licence restrictions, the source of CHTESSEL cannot be made available publicly.

**Author Contributions**

Conceptualization, M.N. and E.D.; numerical simulations and model development: E.D., M.N., C.A., S.B.; data acquisition and processing, F. J., S.E., J.P.M., E.D., M.N.; formal analysis, M.N.; funding acquisition, E.D.; all authors contributed in writing, reviewing and editing the manuscript.

**Competing Interests**

The authors declare no conflict of interest. The funders had no role in the design of the study; in the collection, analyses, or interpretation of data; in the writing of the manuscript, or in the decision to publish the results.





**Table 1.** List of simulations considered in the present study

| Name | Description | Vegetation fraction & type | cvegl & cvegh | LAI |
|------|-------------|---------------------------|---------------|-----|
| ERA5 | ERA5 | IFS | IFS | IFS seasonal cycle |
| CTR | CHTESSEL offline | IFS | IFS | IFS seasonal cycle |
| SFX | SURFEX offline | ECOCLIMAP-II | ECOCLIMAP-II for cvegh and cvegl=1-exp(-0.6LAI) | Interactive |
| H_CCI | CTR with vegetation fraction and types from ESA-CCI | ESA-CCI | IFS | IFS seasonal cycle |
| H_CCI_cl | H_CCI with clumping for cvegl & cvegh | ESA-CCI | cvegh=1-exp(-0.6LAI) cvegl=1-exp(-0.6LAI) | IFS seasonal cycle |
| H_CCI_cl_LAI | H_CCI with clumping for cvegl & cvegh + CGLS LAI | ESA-CCI | 1-exp(-0.6LAI) cvegl=1-exp(-0.6LAI) | CGLS seasonal cycle |


**Table 2.** Bias and RMSE computed from JJA daily maximum LST bias average over entire Iberia, for each model-based dataset considered in the present study, using LSA-SAF LST as reference

| Simulation | Bias [ºC] | RMSE [ºC] |
|------------|-----------|-----------|
| ERA5 | -5.1 | 5.7 |
| CTR | -5.0 | 5.7 |
| SFX | 1.1 | 3.2 |
| H_CCI | -5.8 | 6.2 |
| H_CCI_cl | -4.4 | 5.0 |
| H_CCI_cl_LAI | -1.5 | 3.5 |

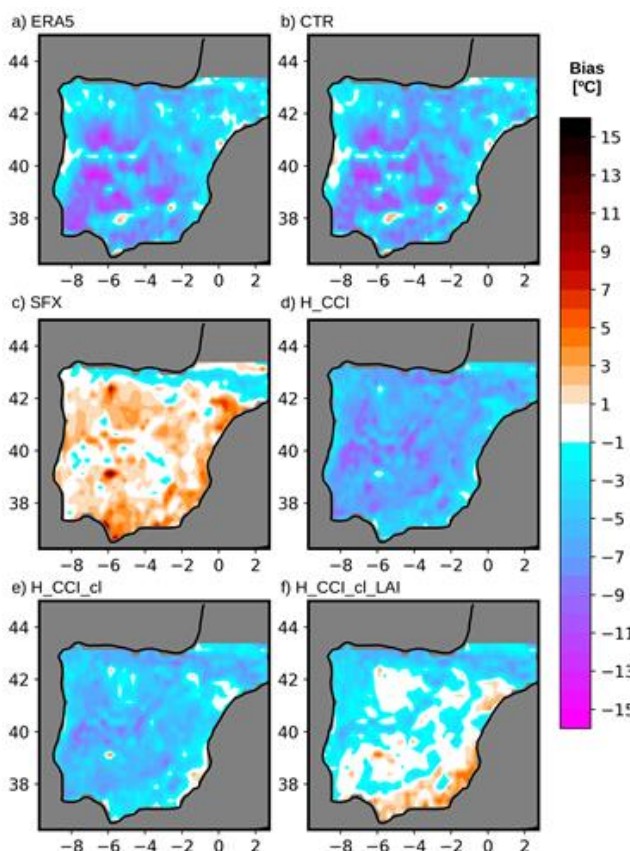

**Figure 1.** Maps of JJA daily maximum LST bias over Iberia under clear-sky conditions, computed for different simulations a) ERA5; b) CHTESSEL offline (CTR); c) SURFEX offline (SFX); d) H_CCI; e) H_CCI_cl; and f) H_CCI_cl_LAI. LSA-SAF LST was considered as reference for computing the simulation errors.





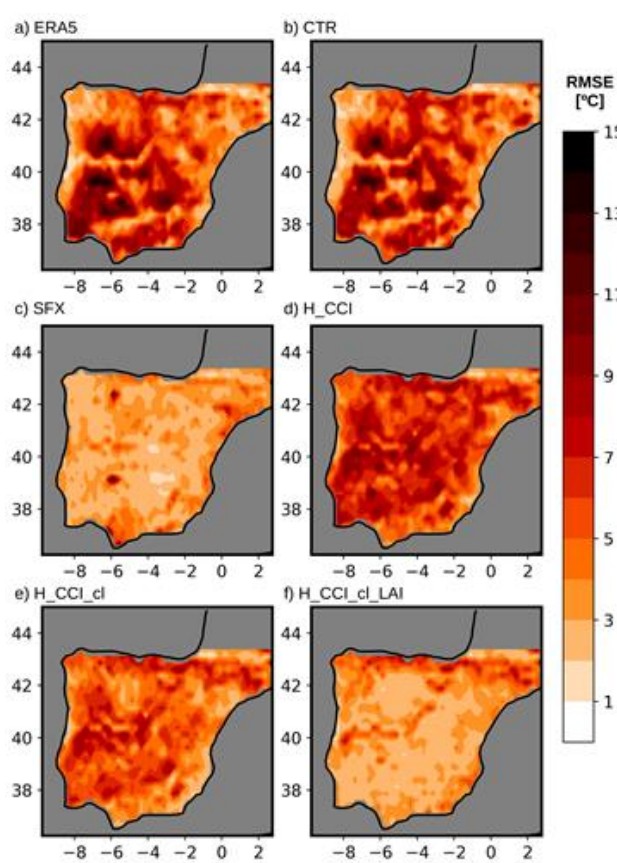

**Figure 2.** Same as Fig. 1 but for RMSE.

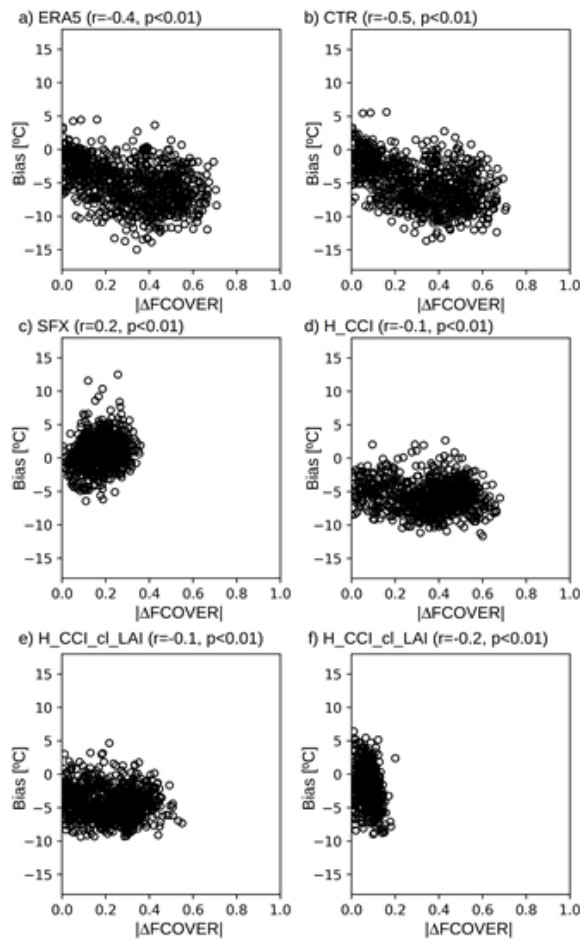

**Figure 3.** Scatter plots of grid point JJA LSTmax bias over Iberia under clear sky conditions against absolute error in monthly maximum FCOVER. a) ERA5; b) CTR; c) SFX; d) H_CCI; e) H_CCI_cl; and f) H_CCI_cl_LAI. LSA-SAF LST was considered as reference for computing the LSTmax errors, while CGLS FCOVER was considered as reference for computing the FCOVER errors. The Pearson correlation coefficient (r) and respective p-value are shown for each simulation.




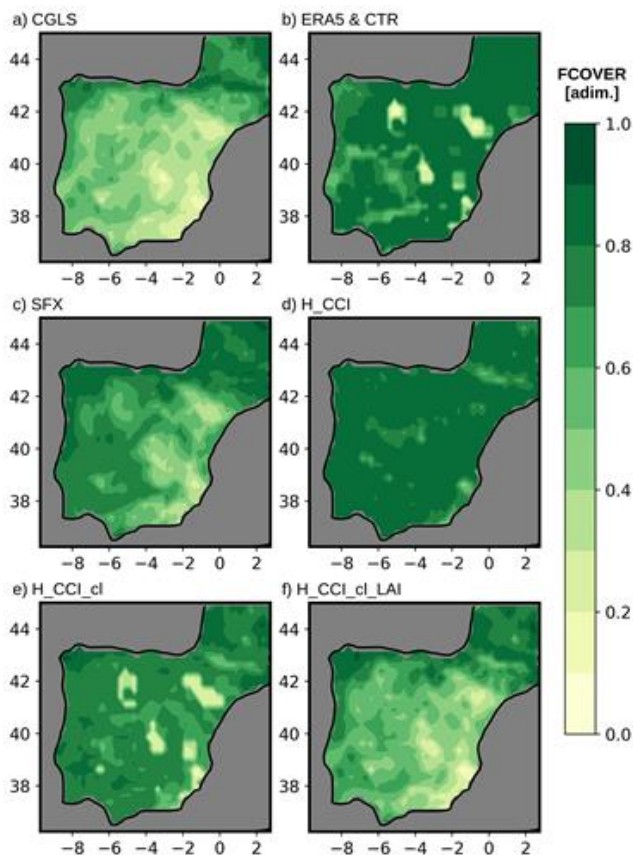


**Figure 4.** Fraction of green vegetation coverage (FCOVER) from a) CGLS dataset; b) ERA5 and CTR (the FCOVER is identical for these two datasets); c) SFX; d) H_CCI; e) H_CCI_cl; and f) H_CCI_cl_LAI.

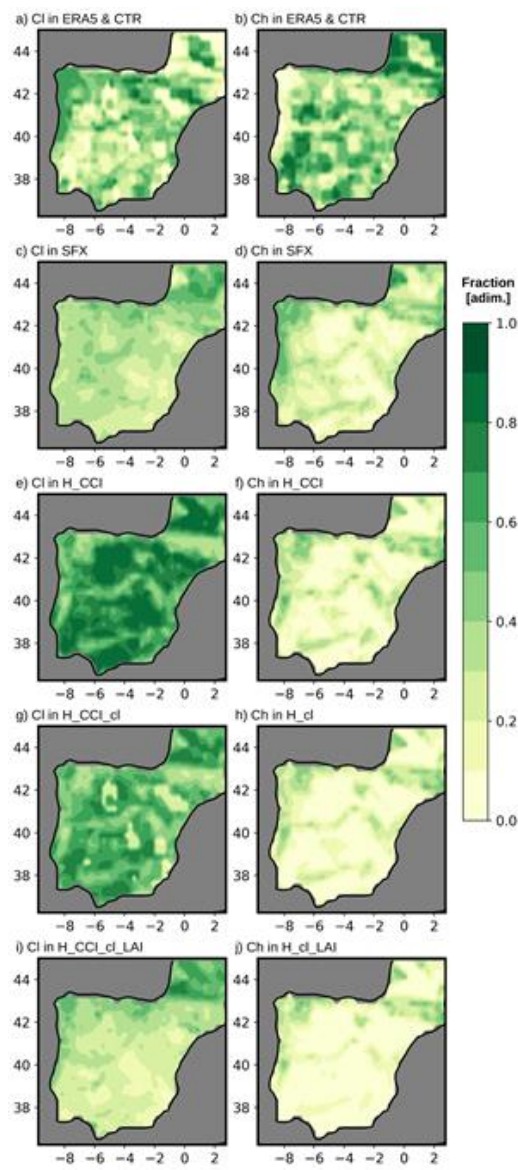


**Figure 5.** Fraction of low (Cl, left column) and high (Ch, right column) vegetation coverage averaged over JJA. From top to bottom the panels represent Cl and Ch in CTR (which is identical to ERA5), SFX, H_CCI, H_CCI_cl and H_CCI_cl_LAI.



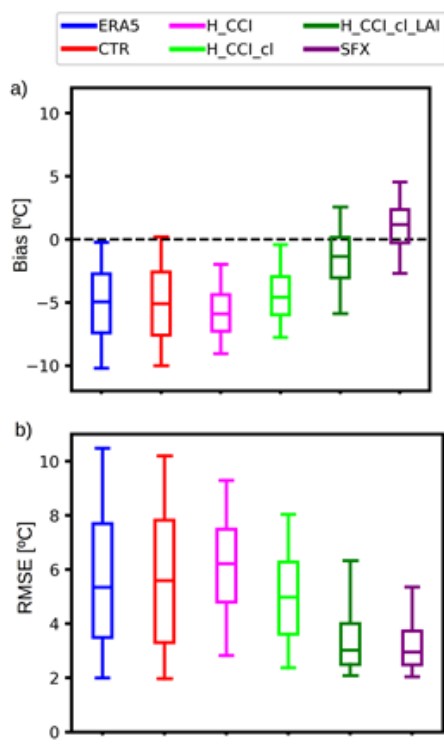


**Figure 6.** Boxplots of JJA daily maximum LST bias (a) and RMSE (b) computed over the 2004-2015 period for from ERA5 (blue), CTR (red), H_CCI (pink), H_CCI_cl (light green), H_CCI_cl_LAI (dark green) and SFX (purple). The boxplot spread represents different grid points over Iberia. Only time instants under clear-sky conditions are considered. The whiskers represent the 5th- and 95-percentiles.






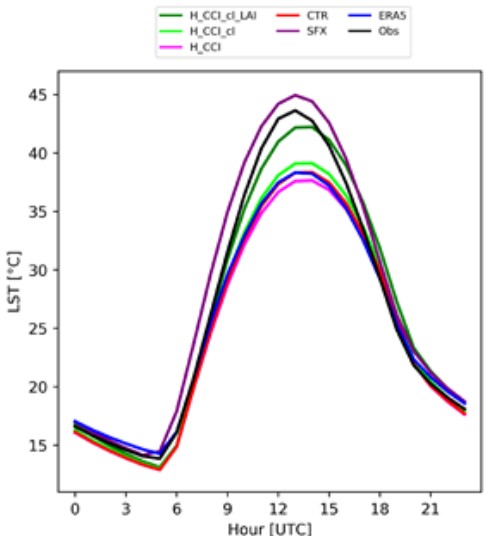

**Figure 7.** Diurnal cycle of JJA LST averaged over all Iberia grid points computed from LSA-SAF (black), ERA5 (blue), SFX (purple), CTR (red), H_CCI (pink), H_CCI_cl (light green), and H_CCI_cl_LAI (dark green). Only time instants under clear-sky conditions were considered.




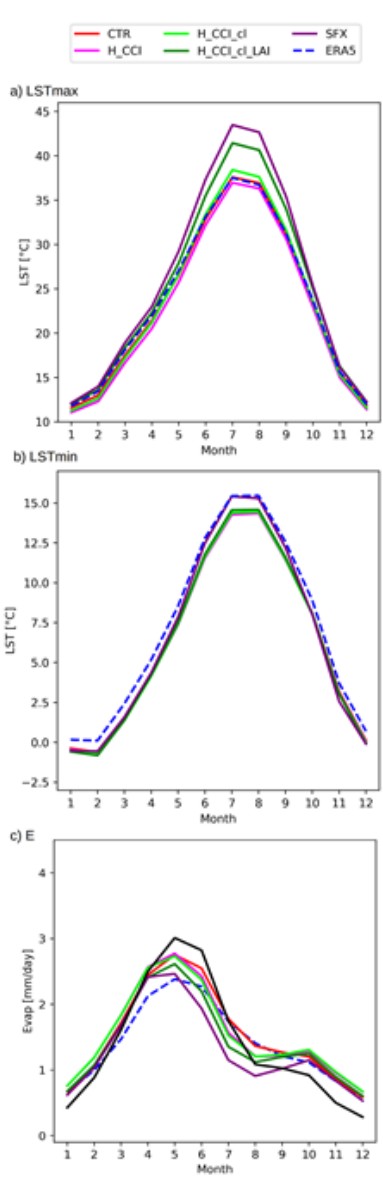

**Figure 8.** Seasonal cycle of a) daily maximum LST; b) daily minimum LST; and c) daily total evaporation. All panels represent averages taken over all Iberia grid points considering all sky conditions, for each month during the 2004-2015 period. Observations are only available for evaporation (GLEAM dataset, black line in panel c), since the availability of LSA-SAF LST is low due to cloud coverage over Iberia outside of summer months.
