# Peer review of "Role of vegetation in representing land surface temperature in the CHTESSEL (CY45R1) and SURFEX-ISBA (v8.1) land surface models: a case study over Iberia"

_Geoscientific Model Development, 2020_

## Referee Comment (RC1) · Anonymous Referee #1 · 14 Apr 2020

Review of "Role of vegetation in representing land surface temperature in the CHT-ESSEL (CY45R1) and SURFEX-ISBA (v8.1) land surface models: a case study over Iberia" by Miguel Nogueira et al.

This study used the LSA-SAF satellite product to identify the summer LST biases in two land surface models (CHTESSEL and SURFEX) over Iberia and then proposed a methodology with the more reasonable vegetation data sets to reduce the large LST cold biases during daytime in CHTESSEL. The offline results have demonstrated the improvement of LST simulation. Some comments are as follows:

[Figure]

1) As the authors mentioned, LST is a key factor in the surface-atmosphere energy and water exchanges. The LST calculation and bias are closely related to the surface fluxes, especially sensible heat flux. This study addresses only LST and evaporation and needs further investigation to demonstrate the propose method also reduced the biases of surface fluxes.

2) It is important to use a coupling system to further address this method, for the land-atmosphere interactions could provide quite different feedback.

3) Global investigation is also required to show its improvement and other impacts.
* * *

---

## Referee Comment (RC2) · Anonymous Referee #2 · 7 May 2020

This paper investigates the impact of the vegetation parameters in the modeling of the land surface temperature (LST) in the CHTESSEL land surface model. It is based on simulations in Iberia during summer, compared to satellite infrared LST estimates and to results from another model (SURFEX). It is triggered by a previous analysis that showed a systematic underestimation of the daily maximum LST by CHTESSEL during summer in Iberia. Different aspects of the vegetation parameterizations are considered and tested (cover fraction, low and high vegetation, LAI). Changes in the vegetation inputs are suggested, with a clumping approach and the addition of seasonality in the

fractional cover. These modifications successfully reduce the LST cold bias, when done in a consistent way among the different vegetation parameters.

The analysis is, for most aspects, carefully conducted, with relevant references to previous works. The paper is well structured and written. It will be an interesting contribution to the field, once the following points are considered.

1) The effects of the vegetation parameter modifications are carefully tested for the daily maximum in summer in Iberia, against IR satellite LST estimates. Different model options are compared for the rest of the year, but are not compared to the satellite LST, on the basis that the cloud cover is too large for the other seasons. The reviewer is fully aware of the difficulty to compare IR LSTs with other estimates, because of possible cloud contamination. However, it is just not legitimate to pretend that the comparisons are impossible outside summer (line 445 and following). Several authors of this paper are directly producing IR LST estimates on a daily basis and the community (including the reviewer) sincerely expects that these estimates are not valid only in summer (especially in Iberia that is not the cloudiest region under the SEVIRI disk). That would cast significant doubts on the usefulness of IR LST to produce CCI records... With increasing cloudiness outside summer, larger uncertainties could be expected, but at the monthly time scale of the analysis, they should not jeopardize the comparison. The authors have to prove that the vegetation modification they propose for the summer period is still valid for the other seasons, in agreement with the observations. It is likely that the results for the rest of the year will be encouraging and it will strengthen the demonstration. It will actually be interesting to discuss the differences in behavior between the clear and cloudy scenes in terms of vegetation impact on the model and their comparisons with the clear sky LST.

2) There seems to be a shift in the diurnal cycle of the summer LST, between the IR LST and SURFEX on one side and the CHTESSEL model on the other side, regardless of the vegetation parameters (Figure 7). The peak in the maximum LST is delayed with the CHTESSEL simulations. Any reasons for that? Any way to correct for it? This

should be commented in the text, even if not corrected.

3) The differences in vegetation parameters from the selected sources are very large (Figures 4-5). Additional comments on the reliability of these datasets, depending on their bases, despite their very 'official' nature? Advices on their applicability for other studies? Some datasets to avoid?

4) Minor points: - Line 56. 8-13 microns, not millimeters. - Lines 72-72. The authors tend to underestimate the uncertainty of the microwave LST estimates and maximize the uncertainty of the IR LST estimates. To be fair, the comparisons have to be done under the same conditions. See Jimenez et al, JGR, 2017 for instance, where comparisons are performed for the same time and same stations: an RMSD of 2.4K is found for the IR (MODIS) and 4.0K for the microwaves (AMSR-E). See also comparisons in Ermida et al., JGR, 2017 between IR estimates and MW estimates. Even between IR estimates the differences can be very large, seriously questioning an uncertainty below 2K for each individual IR product.

---

## Referee Comment (RC3) · Anonymous Referee #3 · 12 May 2020

In this article the authors describe the role of vegetation coverage on the quality of land surface temperature (LST) over Iberia as simulated by reanalysis from ECMWF and by offline integrations of land surface models (LSMs). This is a subject I believe is of interest for readers of Geoscientific Model Development (GMD). Although the conclusions on the impact of vegetation on LST are entirely expected and not surprising, my assessment is that this article provides useful information on the specific products and models it describes (i.e., ERA-5, ERA-Interim, CHTESSEL, SURFEX), and is thus acceptable for publication in GMD.

Major comments:

The introduction is way too long, with too many details that are not really relevant for this specific study.

Throughout the paper, root-mean-square errors (RMSE) are used together with bias. Considering that RMSE includes the effect of bias, which are not negligible in this study, the two evaluation metrics are very closely related and not independant. The authors should rather use the standard deviation of the errors (STDE) (or the unbiased RMSE as called by certain), which is independent from the bias and provides an estimation of the random component of the errors. If the authors do so, the analysis could lead to different conclusions.

Minor comments:

First paragraph of the Introduction: Why not mention the impact of vegetation on evapotranspiration?

Paragraph starting on Line 53 is too long (general comment, too long paragraphs are more difficult to read).

Line 101: Use "estimate" instead of "constrain"?

Line 107: "... land surface model"

Line 212: "Boone and Etchevers..."

Line 213: What is NIT?

Paragraph starting at line 239: be careful with the use of present and past tenses (not consistent in this paragraph, please also verify the rest of the manuscript for inconsistencies).

Line 279: Is using the median for this spatial upscaling the best way? Have you looked at other approaches?
[Figure]

Line 299: "... the dependence of the fraction of green vegetation coverage on the LAI..." and "may render this approximation invalid". Not sure I understand the link between the two statements.

Line 309: "... should be regarded as an additional product rather than the truth..." I would reformulate, as this is valid for every dataset.

Line 331: "... these spatial patterns were tightly related to the corresponding bias patterns". First, should use the present tense. Second, this relation between RMSE and bias is normal, since RMSE includes the effect of bias (see second item in major comments).

Line 346: "Given these results... " This has already been mentioned.

Paragraph starting at line 353: The word "overestimated" is used twice in this paragraph, suggesting an error. The authors should rather say that the values are "larger than" since they are compared with another model product, and not with observational evidence. Same comment for the word "underestimated" at Line 462.

Paragraph starting at Line 486: If you are using Fig. S2 in the main discussion of this article, it should be with the main text, not in the supplement section.

Line 513: "It is important to highlight that, once again, the roughness length change is not effective by itself..." The evidence provided in this article for that statement is anecdotal, and I don't see any reasons why changes to the roughness length is conditional to changes to the changes to other land surface parameters (in general).

Line 525: Change "robust" by "complete". Same for line 528.

Line 555: Gu et al. 2019 is not in the list of references.

Line 578: "By performily ng some additional sensitivity tests, we showed the importance of the patterns of surface roughness lengths of momentum and heat transfer (z0m and z0h) for LST simulated by LSMs". This was only briefly mentioned in the

main text, and evidently not the emphasis of this article. I don't think this should be listed as one of the main conclusions of this work.

Line 581: the roughness length cannot be "observed" (more like an estimation based on an ensemble of measurements).

Line 595 (and other parts of the paper): I think the authors are exagerating the similarity between the offline and 3D simulations. If these study was done in other seasons, the conclusions would have been very different, as evidenced in Fig. 8 which shows that the evaporation with ERA-5 is very different in earlier months from what is obtained with the offline LSM (CTR). Such differences are also found for the minimum LST in the first months of the year.

Line 597: "... hints into the role played by vegetation in land-atmosphere exchanges". This should be removed or rephrased, because this role of vegetation has been known for a very long time, and evidenced by many other studies.

Figure 6: I guess the boxes are for the 25th and 75th percentiles... should this be mentioned in the caption?

---

## Author Comment (AC1) · 4 Jun 2020

We want to thank the Editor and the three Anonymous Reviewers for the constructive comments on our study. Below we provide replies to the all the comments posted by the three Reviewers, along with proposed revisions to three Figures from the manuscript, in order to illustrate some specific points in our replies.

Reviewer # 1

Review of "Role of vegetation in representing land surface temperature in the CHT-

ESSEL (CY45R1) and SURFEX-ISBA (v8.1) land surface models: a case study over Iberia" by Miguel Nogueira et al. This study used the LSA-SAF satellite product to identify the summer LST biases in two land surface models (CHTESSEL and SURFEX) over Iberia and then proposed a methodology with the more reasonable vegetation data sets to reduce the large LST cold biases during daytime in CHTESSEL. The offline results have demonstrated the improvement of LST simulation.

R: We thank the Reviewer for his/her timely and interesting comments, which we believe will help to improve our manuscript. Below we provide point-by-point replies to all the Reviewer's comments.

1) As the authors mentioned, LST is a key factor in the surface-atmosphere energy and water exchanges. The LST calculation and bias are closely related to the surface fluxes, especially sensible heat flux. This study addresses only LST and evaporation and needs further investigation to demonstrate the proposed method also reduced the biases of surface fluxes.

R: We thank the reviewer for raising this point. Over Iberia (our study domain) there are no consistent observational datasets with enough spatial coverage to perform a robust error analysis for the surface sensible heat fluxes. For example, Martens et al. 2020 only considered 1 station from the FLUXNET2015 database over Iberia. Nonetheless, we acknowledge the central importance of surface fluxes, especially sensible heat flux (SH). In this sense we propose to deepen our analysis of the surface fluxes. Specifically, we will include a panel in Fig. 8 with the seasonal cycle of SH in the different offline experiments, and panels in Fig. 7 of surface fluxes diurnal cycle during JJA. We will extend the discussion in Section 6 to deepen the analysis on the impact of the new simulations to the SH and evaporation. (The new figures are at the end of this reply)

Martens, B., Schumacher, D. L., Wouters, H., Muñoz-Sabater, J., Verhoest, N. E. C., and Miralles, D. G.: Evaluating the surface energy partitioning in ERA5, Geosci. Model Dev. Discuss., https://doi.org/10.5194/gmd-2019-315, in review, 2020.

2) It is important to use a coupling system to further address this method, for the land atmosphere interactions could provide quite different feedback. 3) Global investigation is also required to show its improvement and other impacts

R: We understand the importance of the global investigation and of the coupled system. However, both of them are beyond the scope of the present investigation, and their inclusion would make the present investigation too long. In fact, this is the subject of our ongoing research extending what we found in Iberia to the full globe and to coupled simulation performed using ECMWF coupled model. Preliminary results (including coupling) suggest the added value of our proposed changes over Southern Europe and Southern Africa. Our preliminary results also suggest that in some other regions, updating the vegetation coverage requires updating other model parameters beyond just vegetation fraction and density (e.g. handling of snow albedo, surface turbulent exchange coefficients, etc.) Thus, in our opinion, careful regional analyses are required to understand the key processes and sensitivities over different climatic regions, and only after a global solution can be attained. The need for future evaluation using a coupling system and extending to the full globe will be pointed out in the discussion in Section 5, and underlined in the conclusions in Section 6.

Reviewer # 2

This paper investigates the impact of the vegetation parameters in the modeling of the land surface temperature (LST) in the CHTESSEL land surface model. It is based on simulations in Iberia during summer, compared to satellite infrared LST estimates and to results from another model (SURFEX). It is triggered by a previous analysis that showed a systematic underestimation of the daily maximum LST by CHTESSEL during summer in Iberia. Different aspects of the vegetation parameterizations are considered and tested (cover fraction, low and high vegetation, LAI). Changes in the vegetation inputs are suggested, with a clumping approach and the addition of seasonality in the fractional cover. These modifications successfully reduce the LST cold bias, when done in a consistent way among the different vegetation parameters. The analysis is,

for most aspects, carefully conducted, with relevant references to previous works. The paper is well structured and written. It will be an interesting contribution to the field, once the following points are considered.

R: We thank the Reviewer for his/her insightful and constructive comments, which we believe will help to improve the quality of our manuscript significantly. Below we provide point-by-point replies to all the Reviewer's comments.

1) The effects of the vegetation parameter modifications are carefully tested for the daily maximum in summer in Iberia, against IR satellite LST estimates. Different model options are compared for the rest of the year, but are not compared to the satellite LST, on the basis that the cloud cover is too large for the other seasons. The reviewer is fully aware of the difficulty to compare IR LSTs with other estimates, because of possible cloud contamination. However, it is just not legitimate to pretend that the comparisons are impossible outside summer (line 445 and following). Several authors of this paper are directly producing IR LST estimates on a daily basis and the community (including the reviewer) sincerely expects that these estimates are not valid only in summer (especially in Iberia that is not the cloudiest region under the SEVIRI disk). That would cast significant doubts on the usefulness of IR LST to produce CCI records. . . With increasing cloudiness outside summer, larger uncertainties could be expected, but at the monthly time scale of the analysis, they should not jeopardize the comparison. The authors have to prove that the vegetation modification they propose for the summer period is still valid for the other seasons, in agreement with the observations. It is likely that the results for the rest of the year will be encouraging and it will strengthen the demonstration. It will actually be interesting to discuss the differences in behavior between the clear and cloudy scenes in terms of vegetation impact on the model and their comparisons with the clear sky LST.

R: Indeed the satellite - model LST comparison can be performed for any time of the year. However, not only do we have a higher number of model-satellite clear sky matchups during JJA, as it is during that period that ERA5 (and the Control run) shows

the largest biases and higher sensitivity to changes in the vegetation type and cover. Thus, we maintain our focus in JJA. Nonetheless, we acknowledge that completely discarding the LSA-SAF LST information may be exaggerated. Thus, we computed monthly averaged daily maximum and minimum LST over Iberia under clear-sky conditions over the full annual cycle, and included them in Fig. 8 (see end of reply), along with the respective fraction of valid data points. The new results provide further evidence for the neutrality of our vegetation coverage updates outside of the warm months. We also revised the manuscript taking these points into account, with significant changes to Sections 5.2.

2) There seems to be a shift in the diurnal cycle of the summer LST, between the IR LST and SURFEX on one side and the CHTESSEL model on the other side, regardless of the vegetation parameters (Figure 7). The peak in the maximum LST is delayed with the CHTESSEL simulations. Any reasons for that? Any way to correct for it? This should be commented in the text, even if not corrected.

R: In the study of Johanssen et al. 2019 we identified a large shift in the diurnal cycle of the simulations when using 1-hour time-step. This was partially mitigated with the reduction of the time-step to 15min (as in this study). Also in that study, the increased vertical resolution in the soil (changing the top layer from 7 cm to 1 cm) had a positive impact on the diurnal cycle shift. Therefore, this shift is mainly attributed to the numerics of the model, and further attention is required, looking at the time-step (implicit coupling between skin energy balance solver and soil vertical diffusion), vertical discretization of the soil and coupling between skin layer and underlying soil. Although such detailed analysis on the numerics o the model is beyond the scope of the present manuscript, we propose to briefly discuss this issue at end of section 5.1, accounting for the arguments provided in the present reply.

3) The differences in vegetation parameters from the selected sources are very large (Figures 4-5). Additional comments on the reliability of these datasets, depending on their bases, despite their very 'official' nature? Advices on their applicability for other

studies? Some datasets to avoid?

R: This is a very good point raised by the reviewer. We assume that these new datasets (ESA-CCI land cover and CGLS LAI) are the current state of the art in terms of land cover and LAI global mapping, respectively. The study by Fuster et al (2020) includes an assessment of CGLS vegetation products, showing the consistency among different CGLS versions and MODIS LAI. In addition to uncertainties in the mapping of land cover in the dataset, there is also the step of transforming the ESA-CCI land cover to the CHTESSEL dominant vegetation type and associated parameters. From our experience, this last step is crucial and model dependent, and required further attention (ongoing) in revising some of the vegetation dependent parameters, as shown in this study for the case of the roughness length for turbulent heat exchanges.

Minor points:

Line 56. 8-13 microns, not millimeters.

R: Corrected

Lines 72-72. The authors tend to underestimate the uncertainty of the microwave LST estimates and maximize the uncertainty of the IR LST estimates. To be fair, the comparisons have to be done under the same conditions. See Jimenez et al, JGR, 2017 for instance, where comparisons are performed for the same time and same stations: an RMSD of 2.4K is found for the IR (MODIS) and 4.0K for the microwaves (AMSR-E). See also comparisons in Ermida et al., JGR, 2017 between IR estimates and MW estimates. Even between IR estimates the differences can be very large, seriously questioning an uncertainty below 2K for each individual IR product

R: The LSA-SAF SEVIRI LST product has been widely validated using representative stations (e.g. the LSA SAF stations maintained by KIT, lying over plain and highly homogeneous areas, where in situ estimates are representative of the pixel-scale estimates). For these stations, Goettsche et al 2016 reported RMSDs between 1.2 and 1.7

K for Gobabeb, Farm Heimat (Namibia) and Dahra (Senegal); Ermida et al, 2014 reported an RMSD of 1.3 K for SEVIRI at Evora and a RMSD of 3.2 K for MODIS. Indeed several studies indicate higher RMSD values for MODIS, including Jimenez et al. 2017, but validation exercises performed for SEVIRI LST indicate a better accuracy. Given the discrepancy in RMSD values between MODIS and SEVIRI it is to be expected that the two products show significant differences as shown in Ermida et al. 2017. Indeed, the accuracy values presented in line 72 (5-6 K; previous version of the manuscript) for microwave products did not take into account the validation results in Jimenez et al. 2017, while the IR values (1-2 K) referred to the accuracy found for the LSA-SAF IR LST. As such we have edited the text in order to accommodate the validation results in Jimenez et al (2017) and also to take into account a broader view of IR LST products: "MW LST estimates usually have lower spatial resolution, and lower accuracy values, typically in the 4-6 K range (e.g., Aires et al., 2001; Prigent et al., 2016; Duan et al., 2017; Jimenez et al., 2017), when compared with TIR LST, generally within 1-4 K (e.g. Trigo et al., 2011; Goettsche et al 2016; Ermida et al. 2014; Martin et al., 2019)."

Reviewer #3

In this article the authors describe the role of vegetation coverage on the quality of land surface temperature (LST) over Iberia as simulated by reanalysis from ECMWF and by offline integrations of land surface models (LSMs). This is a subject I believe is of interest for readers of Geoscientific Model Development (GMD). Although the conclusions on the impact of vegetation on LST are entirely expected and not surprising, my assessment is that this article provides useful information on the specific products and models it describes (i.e., ERA-5, ERA-Interim, CHTESSEL, SURFEX), and is thus acceptable for publication in GMD.

R: We want to thank the Reviewer for the provided detailed and useful comments, which we believe will help to improve the overall quality of our manuscript. We have addressed all questions, and provide point-to-point replies to all of the Reviewer's comments below.

Major comments:

The introduction is way too long, with too many details that are not really relevant for this specific study.

R: Thank you for your suggestion. We propose to shorten the introduction, particularly reducing the portions concerning the description of MW satellite measurements (in the second paragraph) and concerning the use of satellite observations to constrain model parameters which are not directly linked to vegetation (in the fourth paragraph)

Throughout the paper, root-mean-square errors (RMSE) are used together with bias. Considering that RMSE includes the effect of bias, which are not negligible in this study, the two evaluation metrics are very closely related and not independent. The authors should rather use the standard deviation of the errors (STDE) (or the unbiased RMSE as called by certain), which is independent from the bias and provides an estimation of the random component of the errors. If the authors do so, the analysis could lead to different conclusions.

R: The results from the previous work of Johanssen et al. (2019) suggested that the largest errors in JJA daily maximum LST simulated by CHTESSEL over Iberia corresponded to systematic errors. We have computed the unbiased RMSE which confirms these results, and further shows small differences in the ubRMSE between our different simulations. We propose to add the ubRMSE as Supplementary Information and briefly discuss this point in Section 4.1 (see maps of the ubRMSE at the end of this reply)

Minor comments:

Line 279: Is using the median for this spatial upscaling the best way? Have you looked at other approaches?

R: We also performed the spatial upscaling using the mean and the differences were negligible. The median is more robust to outliers that can occur due to cloud contamination. However, due to the reasonably large amount of pixels used (from 3 km to the 0.25 grid) and screening of at least 70% of valid pixels) the difference between the mean and median were negligible. This information was added in the manuscript.

Paragraph starting on Line 53 is too long (general comment, too long paragraphs are more difficult to read).

R: Paragraph will be shortened, as pointed out above. Line 299: "... the dependence of the fraction of green vegetation coverage on the LAI..." and "may render this approximation invalid". Not sure I understand the link between the two statements.

R:The sentence was removed.

Paragraph starting at Line 486: If you are using Fig. S2 in the main discussion of this article, it should be with the main text, not in the supplement section.

R: Figure S2 was moved to the manuscript. (new Figure 9)

Line 595 (and other parts of the paper): I think the authors are exaggerating the similarity between the offline and 3D simulations. If these study was done in other seasons, the conclusions would have been very different, as evidenced in Fig. 8 which shows that the evaporation with ERA-5 is very different in earlier months from what is obtained with the offline LSM (CTR). Such differences are also found for the minimum LST in the first months of the year.

R: Although we understand the Reviewer's doubts, our preliminary analysis with coupled (nudged simulations) does show a large degree of similarity between the systematic errors in daily maximum LST between offline and coupled simulations, as evidenced in the Fig. 4 posted at the end of the present reply. Thus, our assumption of similarity between offline and coupled is relatively good, but only for daily maximum LST (but indeed, not for evaporation, nor to daily minimum LST). The text will be revised for improved clarity.

Line 513: "It is important to highlight that, once again, the roughness length change is

not effective by itself..." The evidence provided in this article for that statement is anecdotal, and I don't see any reasons why changes to the roughness length is conditional to changes to the changes to other land surface parameters (in general).

R: This sentence was poorly written. We meant that the change to the roughness length by itself was not effective in reducing this particular JJA daily maximum LST bias over Iberia, and not as a general statement. This will be re-written.

Line 578: "By performily ng some additional sensitivity tests, we showed the importance of the patterns of surface roughness lengths of momentum and heat transfer (z0m and z0h) for LST simulated by LSMs". This was only briefly mentioned in the main text, and evidently not the emphasis of this article. I don't think this should be listed as one of the main conclusions of this work.

R: Removed from the main conclusions

Figure 6: I guess the boxes are for the 25th and 75th percentiles... should this be mentioned in the caption?

R: That is correct, this will be added to the caption

Other minor comments

R: All of the following minor comments will be addressed by performing minor corrections to the text

First paragraph of the Introduction: Why not mention the impact of vegetation on evapotranspiration?

Line 101: Use "estimate" instead of "constrain"?

Line 107: "... land surface model"

Line 212: "Boone and Etchevers..."

Line 213: What is NIT?

Paragraph starting at line 239: be careful with the use of present and past tenses (not consistent in this paragraph, please also verify the rest of the manuscript for inconsistencies).

Line 309: "... should be regarded as an additional product rather than the truth..." I would reformulate, as this is valid for every dataset

Line 331: "... these spatial patterns were tightly related to the corresponding bias patterns". First, should use the present tense. Second, this relation between RMSE and bias is normal, since RMSE includes the effect of bias (see second item in major comments).

Line 346: "Given these results... " This has already been mentioned.

Paragraph starting at line 353: The word "overestimated" is used twice in this paragraph, suggesting an error. The authors should rather say that the values are "larger than" since they are compared with another model product, and not with observational evidence.

Same comment for the word "underestimated" at Line 462.

Line 525: Change "robust" by "complete". Same for line 528.

Line 555: Gu et al. 2019 is not in the list of references.

Line 581: the roughness length cannot be "observed" (more like an estimation based on an ensemble of measurements).

Line 597: "... hints into the role played by vegetation in land-atmosphere exchanges". This should be removed or rephrased, because this role of vegetation has been known for a very long time, and evidenced by many other studies.
* * *
[Figure]

**Fig. 1.** Figure 7. Diurnal cycles of JJA a) LST, b) latent heat flux and c) sensible heat flux averaged over all Iberia grid points, computed from LSA-SAF (black), ERA5 (blue), SFX (purple), CTR (red), H_CCI

[Figure]

**Fig. 2.** Figure 8. Seasonal cycles of daily maximum LST under all-sky (a), and clear-sky (b) conditions; daily minimum LST under all-sky (c) and clear-sky (d) conditions; daily total evaporation under all sky

[Figure]

**Fig. 3.** Supplementary Figure 2. Maps of JJA daily maximum LST unbiased RMSE over Iberia under clear-sky conditions, computed for different simulations a) ERA5; b) CHTESSEL offline (CTR); c) SURFEX offline (SF

[Figure]

**Fig. 4.** Figure Daily maximum LST bias under clear-sky conditions computed from ERA5 for period 2004-2015, offline CHTESSEL for period 2004-2015, offline CHTESSEL 2009-2010, and coupled nudged simulation using